# Machine Learning in Injection Molding: An Industry 4.0 Method of Quality Prediction

**DOI:** 10.3390/s22072704

**Published:** 2022-04-01

**Authors:** Richárd Dominik Párizs, Dániel Török, Tatyana Ageyeva, József Gábor Kovács

**Affiliations:** 1Department of Polymer Engineering, Faculty of Mechanical Engineering, Budapest University of Technology and Economics, Műegyetem rkp. 3., H-1111 Budapest, Hungary; parizsrd@pt.bme.hu (R.D.P.); torok@pt.bme.hu (D.T.); ageyevat@pt.bme.hu (T.A.); 2MTA-BME Lendület Lightweight Polymer Composites Research Group, Műegyetem rkp. 3., H-1111 Budapest, Hungary

**Keywords:** injection molding, cavity pressure curve, machine learning, classifiers, quality control

## Abstract

One of the essential requirements of injection molding is to ensure the stable quality of the parts produced. However, numerous processing conditions, which are often interrelated in quite a complex way, make this challenging. Machine learning (ML) algorithms can be the solution, as they work in multidimensional spaces by learning the structure of datasets. In this study, we used four ML algorithms (kNN, naïve Bayes, linear discriminant analysis, and decision tree) and compared their effectiveness in predicting the quality of multi-cavity injection molding. We used pressure-based quality indexes (features) as inputs for the classification algorithms. We proved that all the examined ML algorithms adequately predict quality in injection molding even with very little training data. We found that the decision tree algorithm was the most accurate one, with a computational time of only 8–10 s. The average performance of the decision tree algorithm exceeded 90%, even for very little training data. We also demonstrated that feature selection does not significantly affect the accuracy of the decision tree algorithm.

## 1. Introduction

Injection molding is one of the most widely used plastic processing technologies—more than 30% of plastic products are produced by injection molding [1]. Today injection molding is considered a highly automated mature technology. However, to stay competitive and adapt to the ever-changing market demands, injection molding companies must move towards smart manufacturing, or Industry 4.0 [2]. Industry 4.0 involves the digitization of production, which inevitably leads to the generation of Big Data [3]. The Big Data lifecycle includes the generation, acquisition, storage, processing, and analysis of data. During injection molding, a huge amount of data is generated and can be successfully collected by sensors installed in different units of the injection molding machine and the cavity [4]. Although data is one of the most valuable assets of an intelligent company, many companies have difficulties selecting essential and useful data from the manufacturing process and processing and analyzing this data effectively [5]. Therefore, many efforts have been made to adopt machine learning (ML) techniques for industrial application. Several authors are of the opinion that ML is one of the most important factors in upgrading a traditional manufacturing system to Industry 4.0 [6].

According to Bertolini et al. [6], ML is a set of methodologies and algorithms that can extract knowledge from data and continuously improve their capabilities by learning from experience (i.e., from data accumulating over time). All ML methods can be divided into three groups: supervised learning, unsupervised learning, and reinforcement learning (Figure 1). In supervised learning, the feature acquires the relationship between the inputs and outputs using information contained in the dataset of training examples [7]. All the output data is labeled or grouped. Based on the type of outputs, supervised learning can be divided into two categories: classification and regression. Classification algorithms are used for discrete outputs, while regression algorithms are used for continuous outputs [8]. Unsupervised learning uses unlabeled datasets; therefore its goal is not to make a prediction, but rather to detect or extract patterns in the data, the nature of which may be partially or completely unknown [6]. Reinforcement learning is not concerned with the specific form of the input, but focuses on the action that should be taken under the current state to achieve the final goal [9]. The most explored ML methods are supervised learning, closely followed by unsupervised learning.

One of the main applications of ML algorithms in the injection molding industry is quality management. Product quality in injection molding is quite a complex issue, as quality can be interpreted in various ways [11]. However, three important groups of quality indicators can be distinguished: (1) the stability of dimensions and weight of the produced parts [12], surface properties (roughness, sink marks, weld lines, etc.) [13], and physical properties (mechanical, optical, electrical, etc.) [14]. In most cases, a combination of these criteria is understood as the quality of a part. Nevertheless, several studies have proved that weight is a reliable index characterizing the quality of an injection molded product and process stability, as variation in weight is inversely proportional to part quality [15,16].

Besides the direct measurement of the weight of the part, which is usually a quality control procedure, it is essential to find a reliable process parameter which will allow monitoring and predicting the weight of a part online. Changes in polymer properties, particularly the specific volume of the melt, clearly indicate changes in the weight of the part. Therefore, monitoring the specific volume of melt through controlling the pressure and temperature in a mold cavity is a reliable tool to predict weight variation [17]. According to Zhou et al. [18], the specific volume of melt is mainly affected by pressure. Therefore, they proposed a pressure integral as an effective process parameter to predict product weight variations and characterize the quality of the injection molded parts. The most relevant pressure data come from the runner and mold cavity [19]. The importance of characteristics from pressure curves can have a complex relationship with the quality of products, which may not be described with simple linear functions. In this case, ML could help discover this relation, as ML algorithms work in multidimensional spaces by learning the structure in the dataset [20].

Recent progress in the application of ML in injection molding is summarized by Selvaray et al. [21]. Zhao et al. [22] used the ML approach to optimize the processing parameters to achieve a target weight of an injection-molded product. The authors used the support vector machine (SVM) method together with the particle swarm optimization (PSO) algorithm. The proposed ML approach enabled stable injection molding. The deviation of product weight was only 0.0212%. Yin et al. [23] proposed a back-propagation neural network to predict and optimize the warpage of injection-molded parts based on the main process variables, including mold temperature, melt temperature, packing pressure, packing time, and cooling time. The proposed method was able to predict the warpage of injection-molded parts within an error range of 2%. Ogorodnyk et al. [24] applied four ML methods to create prediction models for the thickness and width of the injection-molded HDPE tensile specimens based on the injection molding process parameters. The authors found the best correlation coefficient was achieved with the random forest algorithm, while the second-best results were produced by the multilayer perceptron (MLP) neural network method. The reduced error pruning decision tree (REPTree) performs slightly better than the k-nearest neighbor (kNN) algorithm. The authors concluded that overall, all the four methods showed good prediction capabilities. Ke and Huang [25] used an MLP neural network to evaluate the quality of injection-molded parts. Instead of using processing parameter settings as inputs, the authors used the so-called “quality indices” [26] extracted from the system and cavity pressure curves. The maximum achieved accuracy of the proposed prediction method was 94%. Gülçür and Whiteside [27] also used quality indexes connected with cavity and system pressure and the position of the injection piston to predict the quality of micro-injection-molded parts. They used a linear regression model, which predicted quality with an accuracy of 84%.

Several authors confirmed that cavity pressure is a valuable data source that can represent the quality of injection molded products [27,28,29]. In a typical cavity pressure profile, several feature points can be extracted that define the characteristics of the injection molding conditions [28,29]. However, the question still remains: which and how many features should be selected from the cavity pressure profile to adequately characterize and predict the quality of injection-molded products? For example, Huang et al. [30] used four features from the cavity pressure curve: the peak pressure, the pressure gradient, the viscosity index, and the energy index. Gim and Rhee [28] proposed five features to be extracted from cavity pressure profiles: the starting point of the filling stage, the switchover point from filling to packing, maximum cavity pressure, the endpoint of the packing stage, and end of the cooling stage. Determining the optimal number of features is a difficult question. However, Hua et al. [31] give some recommendations on how to define the optimal number of features based on sample size. The general rule is that the sample size must exceed the number of features [31]. For example, for the LDA algorithm and a sample size of 30, the optimal number of features can vary from 3 to 12, depending on the correlation of the features. Jain and Waller [32] claim that the optimal feature size is proportional to √n, where n is sample size. One more rule that can help to define the optimal number of features is the Vapnik–Chervonenkis (VC) inequality [33], which gives an upper bound for generalization [34]. However, this generalization rule is only true if the VC dimension is finite, which is not the case for the kNN algorithm, for example, with k = 1 [35,36]. For discrete classifiers, there is a more accurate approach, which recommends a significantly smaller size of the learning sample [37]. In summary, the optimal number of features can differ for different classification algorithms.

Many authors confirm that classification and regression ML algorithms can predict and control the quality of injection molding well. However, the great variety of the ML algorithms and the individual features of each production run requires the development of a new prediction method. In this study, we aim to compare the accuracy and effectiveness of four classification algorithms in predicting the quality of multi-cavity injection molding. We used pressure-based quality indexes as inputs for the classification algorithms.

## 2. Materials and Methods

### 2.1. Experimental Setup

For our experiments, we used an Arburg Allrounder 420 C 1000-290 injection molding machine (Arburg GmbH+Co., Loßburg, Germany) to make products from acrylonitrile butadiene styrene (ABS), named Terluran GP-35 (INEOS Styrolution, Manchester, United Kingdom). The injection molding machine has a distance of 420 mm between the tie bars (420 mm × 420 mm), and the maximum clamping force is 1000 kN. The screw of the injection unit has a diameter of 30 mm with an effective screw length of 23.3 (L/D); the maximum shot volume is 106 cm^3^, the maximum injection pressure is 2500 bar, and the maximum screw torque is 320 Nm. The mold was tempered with a Wittmann Tempro Plus 2 90 (Wittmann Technology GmbH, Vienna, Austria) mold temperature control unit with a maximum pump capacity of 60 L/min. The specimens were injection molded in a 16-cavity mold, which contains 30 pressure sensors (Cavity Eye Ltd., Kecskemét, Hungary) along the flow paths and in different cavities. These sensors were PC15-1-AA indirect pressure sensors and were installed under the ejector pins. Eight sensors were mounted into the fixed side of the mold and the rest were located in the movable side of the mold. Figure 2 shows the layouts of the cavities (a) and the positions of the sensors in the mold (Figure 2a,b). The products were 15 mm × 15 mm square flat specimens with a wall thickness of 1.5 mm. In industrial production, changes in the material batch, drying, or various manufacturing defects can change the shape of pressure curves, so we used settings that achieve similar effects. To produce parts with different masses, we changed the holding phase in our experiments. We used three holding pressures—200 bar, 600 bar, and 1000 bar—with a varied holding time from 0 s to 3 s in 0.25 s steps. The other settings were not changed during production (Table 1). We took five samples produced with each setting after the process became stable and measured the masses of the parts with an Ohaus Explorer analytical balance (OHAUS Europe GmbH, Uster, Switzerland). We also recorded the in-mold pressure data for 10 s from the start of the injection phase, with a sampling rate of 100 Hz. With a holding time of 0 s, we injection molded two series to examine the repeatability of the injection molding machine; thus, had a total of 190 specimens. For the classification procedure and data analysis, we used the MATLAB R2021a platform (The MathWorks Inc., Natick, MA, USA).

### 2.2. Methods

#### 2.2.1. Preparation of Data

This paper aimed to show how simple classifiers can be used to predict the quality of injection molded products from pressure measurements. We examined four classifiers for quality prediction: k-nearest neighbor (kNN), naïve Bayes, binary decision tree, and linear discriminant analysis (LDA). Based on the literature review, we used the products and the pressure curves, and only used the data from the first cavity and its runner (Figure 2b).

We first defined 19 features from cavity pressure curves in this study. Then we implemented a feature selection procedure to determine the optimal set of features for each of the examined classifiers. We defined nine features for both the postgate (*PG*) and the end-of-cavity sensors (EOC), which gives a total of 18 features from the pressure curves (see Equations from (1) to (9)). The 19th feature was defined only from the *PG* sensor and the pregate sensor (*PRG*) (see Equation (10)).
(1)$$PI_{PG}=\int_{0}^{10}P\left(t\right)\;dt$$
(2)$$PI_{PG,t\left(0-P_{max}\right)}=\int_{0}^{t_{max}}P\left(t\right)\;dt$$
(3)$$PI_{PG,t\left(0-P_{max}\right)}/PI_{PG,t\left(P_{max}-10\right)}=\frac{\int_{0}^{t_{max}}P\left(t\right)\;dt}{\int_{t_{max}}^{10}P\left(t\right)\;dt}$$
(4)$$P_{PG,max}=\max\left(P\left(t\right)\right)$$
(5)$$t_{PG,P_{max}}=\arg\underset{t\in \left[0,10\right]}{\max}\left(P\left(t\right)\right)$$
(6)$$t_{PG,first}=\;\min\{t|P\left(t\right)\;\geq \;5\}$$
(7)$$PD_{PG,first}=\frac{P\left(t_{PG,first}+0.01\;s\right)-P\left(t_{PG,first}\right)}{0.01\;s}$$
(8)$$PD_{PG,P_{max}-}=\frac{P\left(t_{PG,P_{max}}-0.09\;s\right)-P\left(t_{PG,P_{max}}-0.1\;s\right)}{0.01\;s}$$
(9)$$PD_{PG,P_{max}+}=\frac{P\left(t_{PG,P_{max}}+0.1\;s\right)-P\left(t_{PG,P_{max}}+0.09\;s\right)}{0.01\;s}$$
(10)$$\Delta PI_{gate}=PI_{PRG}-PI_{PG}$$

PI stands for the integral of the pressure curves, and the lower index always shows which sensor position the pressure curve was obtained from. An additional subscript may indicate the time or time interval from where the feature was calculated. When there was no time interval, the whole pressure curve was used, which means 10 s after the beginning of the injection phase. The notation PD indicates the numerical differentiation of the pressure curve. P and t indicate pressure and time, respectively. The pressure integral values are related to the pressure characteristics for the defined intervals [38]. The ratio of the integrals shows the ratio of the pressure rise to the pressure fall. The different time values give an overall view of the injection and compression phase [38]. The derivatives show the change of the rate of pressure at these time instants. The integral change on the gate provides information about gate freezing and the pressure drop on the gate. The values of these features were then standardized, so the dataset for each feature has a mean of 0 and a standard deviation of 1. For clarity, most of the features (except (1), (3) and (10)) are illustrated in Figure 3.

All the specimens produced were grouped into three classes (11) based on their weight:(11)Undercompensated < Acceptable < Overcompensated

Parts between 0.470 and 0.475 g were classified as “acceptable”. A narrow range was chosen as acceptable, so the number of acceptable products was low compared to the number of non-acceptable products. Typically, for such a small product, a narrow range of acceptable values would be expected in the industry; in our case, it was slightly more than 1% of the weight. The distribution of the weights of samples is approximated with the kernel function. We also highlighted the probability of random sampling from the acceptable class (Figure 4).

#### 2.2.2. Random Sampling

The results of mass measurements were collected in a table with 190 rows and 20 columns (19 columns for features and 1 column for the class of the sample). From this table, different training sets were created by random sampling.

We investigated how well these classifiers work with our data in the case of small datasets (less than ten training samples per each class). For this, we first randomly selected training samples from each class. Then we stored the index of the selected samples in a new table (index table). Note that indices show the place of the training samples in the original dataset. Then we repeated this process a hundred times to investigate the robustness of the training process. This whole sampling process was repeated with 2, 3…10 training samples per class (Figure 5). Each cell in the index table contains as many indices as we want to use for training. If a cell from the second column of the table is selected, it means that we will use two samples per class, so there will be six indices in the cell in total. For each classification, one cell is selected, which gives information about how many and specifically which data are used for training. This index table allowed the different classification algorithms to be tested on identical sets of samples.

#### 2.2.3. Formulating of Feature Datasets

Another goal was to investigate which selected features are the most important for the classification of the mass of parts. We used three feature sets for our research, which were determined from the original dataset. The first feature set was the original one, which contained all the features from the original dataset, so it had 19 columns and was referred to as a complete dataset (CD). The second dataset was generated with principal component analysis and was referred to as the modified dataset (MD). We analyzed the CD to obtain the directions that explain the variance of data. We used the first eight principal components as new dimensions for our data because these components explained more than 95% of the variance. The third feature set was defined separately for each classifier with a selective forward feature selection method (SFS) and was referred to as the selected dataset (SD). The SFS used a leave-one-out cross-validation method with the CD and selected the features for each classifier that were best suited for the classification. Thus, the dimensions in the SD may differ for each classifier, but the generation method was the same. We used each dataset type for all classification methods, and the whole process can be seen in Figure 6.

#### 2.2.4. Classification Algorithms

The kNN algorithms were used with default settings, which means the algorithm searched for the first nearest neighbor, the distance metric was Euclidean distance, and there was no distance weighting. The naïve Bayes classifier used a kernel distribution instead of a normal distribution for fitting; the kernel smoother type was Gaussian, and prior probabilities were calculated from the relative frequencies of the classes from training data. The decision tree classifier used the default settings except the minimum number of branch node observations parameter, because the default size is 10, which would not allow the tree to branch in the case of 1 or 2 training data per class. Therefore, we changed this parameter to the actual number of training samples per class. The LDA classifier used most of the default settings, except that all classes had the same diagonal covariance matrix.

#### 2.2.5. Method of Comparison of Classifiers

After the training of a classifier, the performance of each algorithm was estimated with the accuracy, which was calculated as follows (12).
(12)$$Accuracy=\frac{Number\;of\;correct\;classification}{Number\;of\;all\;classification}\cdot 100$$

The results are presented in a usual plot with mean and standard deviation (Figure 7a) and with a boxplot (Figure 7b), which gives more information about the distribution of the results. In each case of a cardinal number of training data, there were 100 different classifications with training data determined from the index table, and the remaining data were used for testing. The boxplot suggests that the distribution of accuracy is non-Gaussian. To prove this, we used two normality tests with each cardinal number of training samples: the Shapiro–Wilk test and the Kolmogorov–Smirnov test. The null hypothesis of both tests is that the accuracy results come from a normal distribution at a 5% significance level, but they approach the calculation with a different method. On the sample size of 100, the power of the Shapiro–Wilk test is mostly greater than that of the Kolmogorov–Smirnov test. Still, Razali and Wah [27] showed that the skewness and kurtosis coefficients of distribution have a significant effect on the power of these tests. Therefore, distributions were only considered normal if both tests showed that we cannot reject the null hypothesis.

## 3. Results

### 3.1. The KNN Classifier

First, we used the CD with all the features derived from the pressure curves. The results (Figure 7a) indicate that with the increase of the number of training samples, the mean of accuracy increased as well. Standard deviation decreased until the curve reached a plateau. The boxplot (Figure 7b) shows that the interquartile region and the difference between the minimum and maximum values decreased dramatically with more training data. The results of the MD (Figure 8), the dataset with principal components, did not show much difference compared to the results of the CD. However, the SD had a great effect on the accuracy of the classifier; this dataset contained the following features: *PI_PG_*, *PI_PG,t_*_(0*−Pmax*)_, and Δ*PI_gate_*. We showed with the normality test that the results of classifications came from a non-Gaussian distribution. Therefore, we performed a Kruskal–Wallis one-way analysis of variance for each cardinal number of training data, with a significance level of 0.05 to show a difference between the results of classifications. The null hypothesis (H_0_) of the test is that the mean ranks of the groups are the same. We can reject the null hypothesis if the calculated *p*-value is smaller than the significance level. We estimated the epsilon squared estimate of effect size (13) for the Kruskal–Wallis test (Table A1 in the Appendix A).
(13)$$\varepsilon_{r}^{2}=\frac{\chi^{2}}{\left(n^{2}-1\right)/\left(n+1\right)},$$

In Equation (12), $\varepsilon_{r}^{2}$ means the estimate of effect size, $\chi^{2}$ corresponds to the $\chi^{2}$-statistic, and n is the total number of classifications. If epsilon squared is 1, it means perfect correlation, and if it is 0, it indicates no relationship [28]. The analysis showed that there is a significant difference between the classifications. Therefore, we performed the Dunn–Šidak post hoc test, which makes a pairwise comparison of each tested group and controls the Type I error rate (H_0_ is rejected, but it is true) with adjusted *p*-values (*p**). If the adjusted *p*-value is smaller than 0.05, we can conclude that there is a significant difference between the groups. When all features (CD) and principal components (MD) were used, we could not reject the null hypothesis (Table A1 in the Appendix A). However, the feature selected by SFS made a significant difference compared to CD and MD (Figure 8).

### 3.2. Naïve Bayes Classifier

We used a naïve Bayes classifier for our research with the three datasets mentioned earlier. The SD for this classifier contained only one feature (*PI_PG_*). In the case of two training data per class, we observed a drastic average performance drop when the feature set was defined from principal components (MD) or the SFS method (SD) (see Figure 9). This is due to the distribution of the training data. In the case of one training point per class, the spreads of the kernel distributions calculated from the samples are similar for each category. However, when we had two or more samples per class (for training), there were cases when the classification was particularly inaccurate due to the random sampling and the distribution of the original dataset. In certain cases, the training samples come from a very small environment for some classes, while for other groups, the distance between the samples can be several times larger. In these cases, the kernel distribution fitted for the first type of data will be very tight, while a much wider distribution will be fitted on the data with a large distance. Because of the width of the wider distribution, the data far from its training data will more likely belong to that distribution, even near the training samples of the other class. The more training samples there are, the better the algorithm recognizes the distribution of the classes. The normality test proved that the results of the classification did not come from a normal distribution. The Kruskal–Wallis test confirmed that the different feature sets caused, in most cases, significant changes in the accuracy of classifiers (Table A2 in the Appendix A). In the case of one training data per class, the post hoc test showed no significant difference between the classification results when CD was used and the classification results from MD. The use of SD did not make a large difference in some cases where CD was used.

### 3.3. Decision Tree Classifier

We made classifications with decision tree classifiers with the datasets. According to the normality tests, the accuracy results from the classifications did not have a normal distribution. The post hoc tests showed that selective feature selection had no significant effect on the accuracy of the classifier compared to the case when the classifier used all features (see Figure 10a and Table A3 in the Appendix A). Still, the calculation time for SFS was three times more than classification time, so it is preferable not to use feature selection for this classifier. The SD dataset contained only one feature (*PI_PG_*) for the decision tree classifier.

The reason for the similar results of CD and SD was that the decision tree does not always use each feature for classification. It chooses dimensions with which it can make splits and new nodes (Figure 10b). The training of the decision tree is based on the Gini index (14), which showed the impurity of the nodes in the decision tree.
(14)$$G=1-\sum_{i=1}^{c}p_{i}^{2}\;,$$
where G is the Gini index; $p_{i}$ is the proportion of the observations in the *i-*th class in the node compared to the number of total observations.

For each node, node risk can be calculated with the following Equation (15).
(15)$$R_{j}=P_{j}\cdot G_{j}$$
where $R_{j}$ is the so-called node risk of the *j-*th node, $G_{j}$ is the Gini index calculated for the *j*-th node, and $P_{j}=m_{j}/m_{all}$ where $m_{j}$ is the number of data in the *j*-th node, and $m_{all}$ denotes all data in the tree during training.

The growth of importance on the selected node of the feature can be calculated for each node with Equation (16).
(16)$$\Delta I_{j}=\frac{R_{j}-R_{j,1}-R_{j,2}}{N_{split}}$$
where $\Delta I_{j}$ is the growth of the importance of a feature on the *j-*th node, $R_{j}$ is the risk of the parent node, $R_{j,1}$ and $R_{j,2}$ are the risk of the children nodes of *j*-th node, and $N_{split}$ denotes the total number of branches in the tree.

The overall importance of a feature can be calculated by summing the growth of importance for that feature (17).
(17)$$I_{f}=\sum_{j=1}^{N}\Delta I_{j,f},$$
where $I_{f}$ is the importance of the selected feature f, $\Delta I_{j,f}$ is the growth of importance for the selected feature f on the *j*-th node, and N is the number of nodes in the decision tree. If a node does not use features (such as leaf nodes) or does not use the selected feature, then the growth of importance on the selected node is zero.

The similar accuracy results with the CD and with the SD are possible due to the similar way they build their trees. Therefore, we examined the feature importance values when the model could use all features. The results show that the classifier with CD sometimes uses several features to build the tree, such as $PI_{EOC}$, but the importance of $PI_{PG}$ is ten times greater than the importance of the other features. This means that the splitting made by $PI_{PG}$ is more significant than the others. The information carried by the other features has no significant effect on the accuracy in this case if used at all.

### 3.4. Discriminant Analysis Classifier

We used the discriminant analysis classifier, but in this case, training with one sample per class was not possible because the classifier needed more data points than the number of classes to calculate the covariance matrix. The SD increased the accuracy of the classifier in each case (Figure 11) significantly (Table A4 in the Appendix A). For the discriminant analysis classifier, the SD contained the following features: *PI_EOC_*, *P_PG,max_*, and *PI_PG_*_,*t*(0−*Pmax*)_/*PI_PG_*_,*t*__(*Pmax*−10)_.

This was because the classes sometimes overlap, which makes classification more difficult. In addition, many features are more likely to have outliers along a dimension, and these outliers shift the mean of the normal distribution along that dimension with few samples. Figure 12 shows an example of the new test data, which originally belong to the “Undercompensated” class. However, from the contour plots of normal distributions (Figure 12a), it looks as if this test sample should come from the “Acceptable” class. Classification is easier when we show these samples on a different plane (Figure 12b). In this example, the classifier with only features selected by SFS had 11% higher accuracy. When there are many such dimensions with significant overlaps, the accuracy of the classifier is significantly impaired.

When the classification was made with MD, the more components were used, the less information they added to the model, and therefore the more overlap there was between classes. The subspace defined with the sixth and seventh principal components cannot recognize the classes correctly, especially for the test data that originally came from the incomplete class (Figure 13a). If the first and second principal components were used, the classification for this special case (Figure 13b) would be much more accurate.

### 3.5. Comparison of the Most Accurate Classifiers

The comparison of feature sets showed that the feature sets defined with the SFS method (SD) were more efficient for each classifier than the features defined from principal components (MD) or when all the features were used (CD). The effectiveness of classifiers was also compared (Figure 14) for the case when the feature set with the best performance was used. The results (Table A5 in the Appendix A) show that the decision tree classifier would be the best choice; however, for more (but still few) training data, each classifier would predict the class of the injection molded samples quite well. After the statistical analysis, we showed that the decision tree and naïve Bayes classifier results in the case of one sample per class were the same with the effect size of zero. In this case, the boundaries of decision trees are the same as boundaries defined from posterior probabilities.

The computation time of classification (Figure 15) for the naïve Bayes classifier was the longest (117 s), and the other three classifiers processed the data much faster (8–10 s). The computation time of SFS was different for each classifier; the decision tree and naïve Bayes classifier needed the shortest time (34–35 s). In the case of kNN, the selection of features required more time (64 s). The most time was required when SFS worked with the LDA classifier (101 s). Even though the classifications were almost equally accurate for ten samples per class, the calculation time of classification and feature selection made the decision tree the most favorable classifier in this case (Table 2).

## 4. Conclusions

We proved that simple classifiers can be used to predict quality in injection molding even with few training data. For our research, we collected the pressure data directly from the mold cavity. We defined the most important features for each classifier from the pressure curve with a selective feature selection method and principal component analysis. We found that the feature selection method delivers better results than the principal component analysis method. Moreover, the accuracy of classification when the features were defined with the principal component analysis was sometimes worse than when the whole dataset was used. Selective forward feature selection significantly improved the accuracy of predicting the quality of parts.

We compared the accuracy of four different classifiers with different training data sizes, and proved that with the same training dataset, the decision tree algorithm was the most accurate, with a short computational time (8–10 s). The average performance of the decision tree algorithm was more than 90% in every case, even when we used a very low number of training data (1–3). Feature selection does not affect the accuracy of the decision tree algorithm significantly. The second best performance was demonstrated by the linear discriminant analysis algorithm. The naïve Bayes classifier, in the case of a higher number of training samples, showed the same or slightly better accuracy than LDA. At the same time, naïve Bayes did not show good results in the case of a small number of training data. The principal component analysis method has a negative effect on the accuracy of the LDA and the naïve Bayes algorithm. The kNN algorithm can reach the same accuracy as the naïve Bayes classifier and is not very sensitive to the number of training data. In the case of kNN, the feature selection method improved accuracy, while the principal component analysis had no significant effect on it. The computational time of training and classification for naïve Bayes algorithms was the highest out of all the examined algorithms. We proved that it is possible to train simple classifiers with a small amount of training data and reach a high accuracy of prediction.

## Figures and Tables

**Figure 1 sensors-22-02704-f001:**
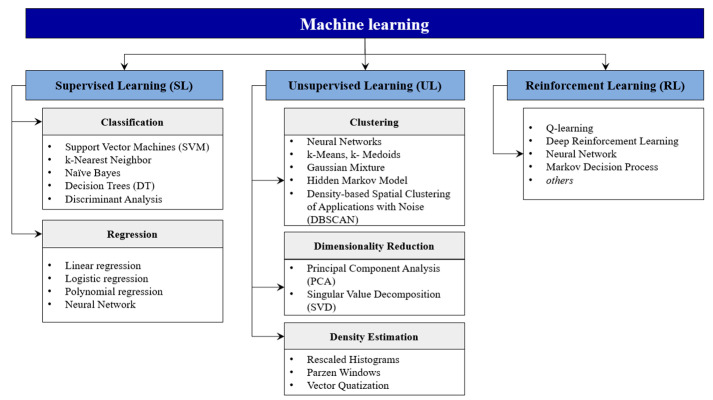
The classification of ML algorithms (based on [6,10]).

**Figure 2 sensors-22-02704-f002:**
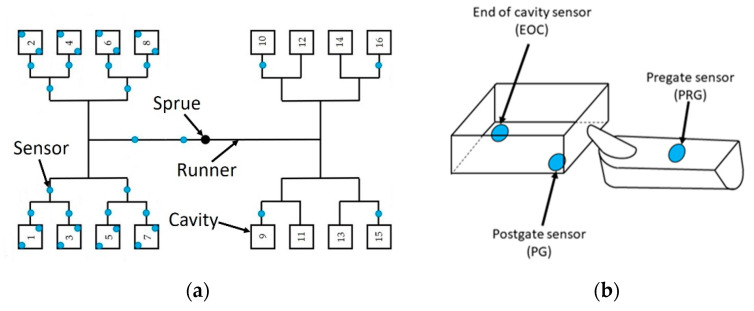
Injection mold and the pressure sensor (blue dots) positions: (**a**) general scheme of the runner system with cavities; (**b**) sensor locations and their names in the first cavity.

**Figure 3 sensors-22-02704-f003:**
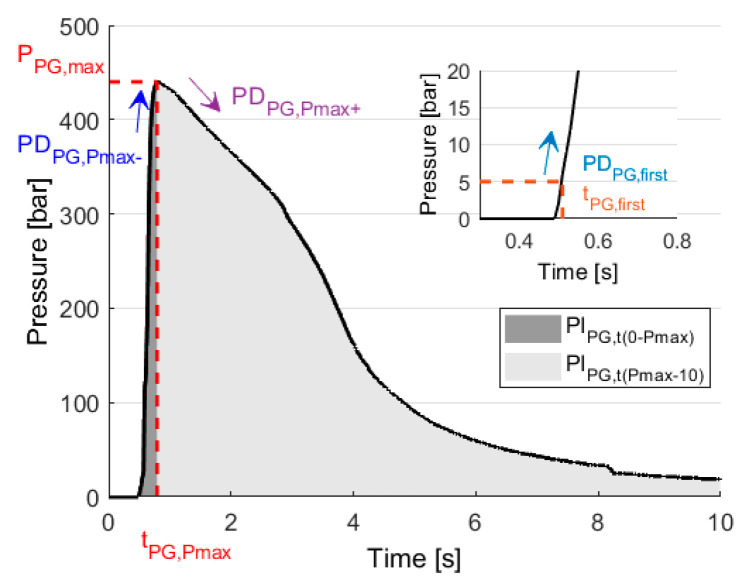
Illustration of the features extracted from the pressure curves.

**Figure 4 sensors-22-02704-f004:**
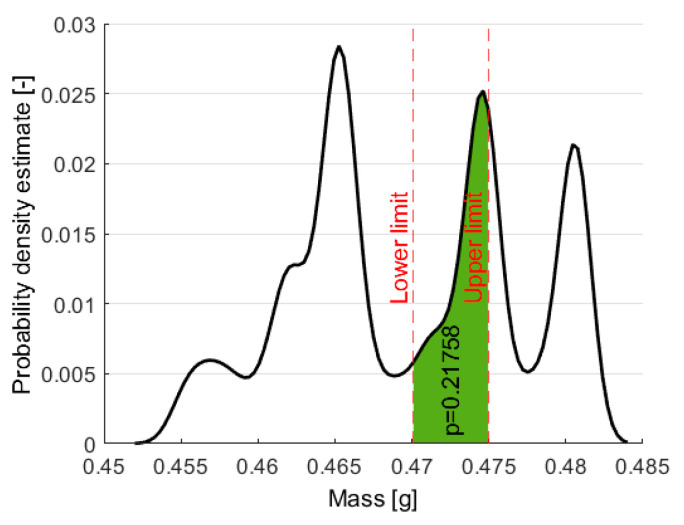
The probability density function of masses calculated with the kernel function (and the *p*-value, which shows the probability of random sampling from this class).

**Figure 5 sensors-22-02704-f005:**
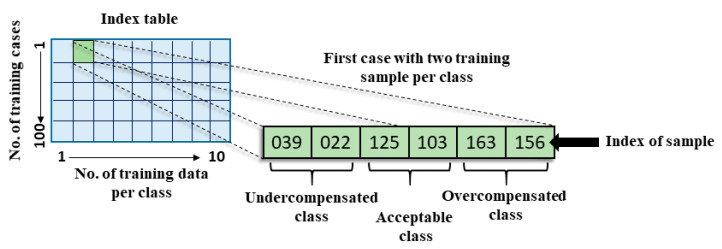
The index table.

**Figure 6 sensors-22-02704-f006:**
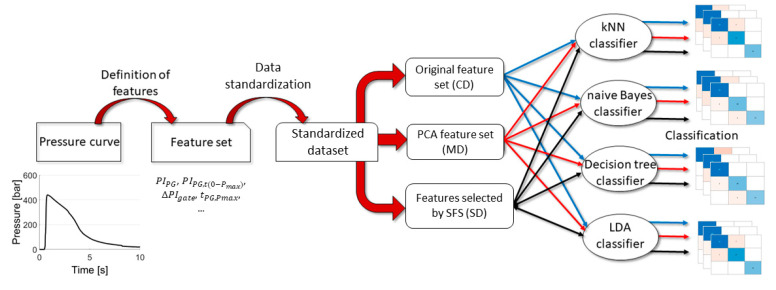
The schematic representation of the classification process.

**Figure 7 sensors-22-02704-f007:**
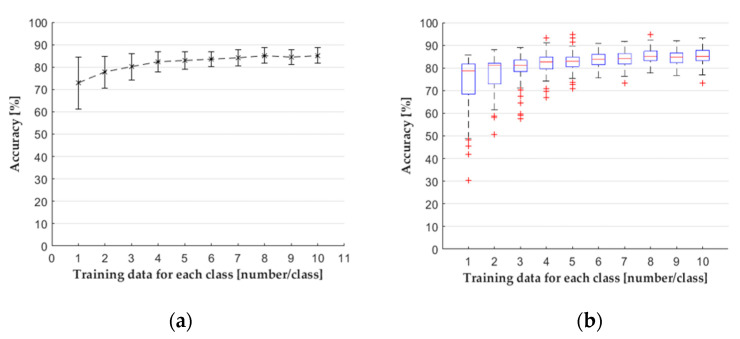
The performance of the kNN classifier when all features were used (CD) showed with (**a**) mean and standard deviation; (**b**) boxplot.

**Figure 8 sensors-22-02704-f008:**
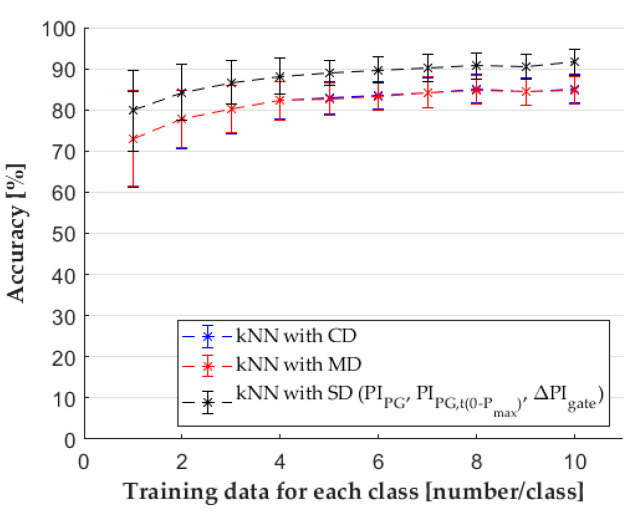
The results of classification with different feature sets.

**Figure 9 sensors-22-02704-f009:**
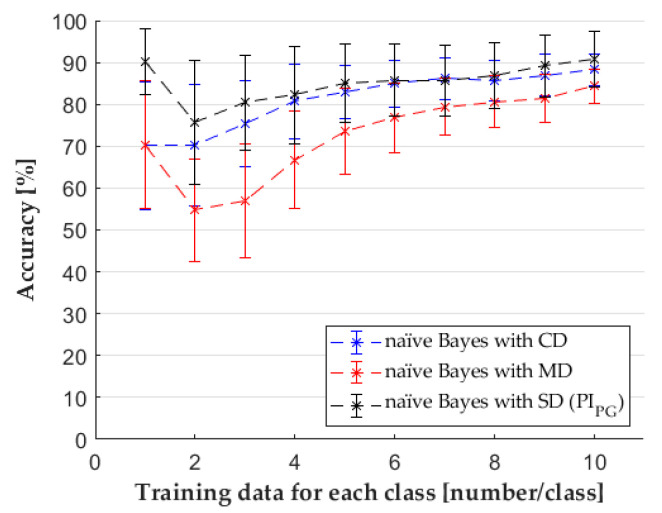
The results of naïve Bayes classifier with the different datasets.

**Figure 10 sensors-22-02704-f010:**
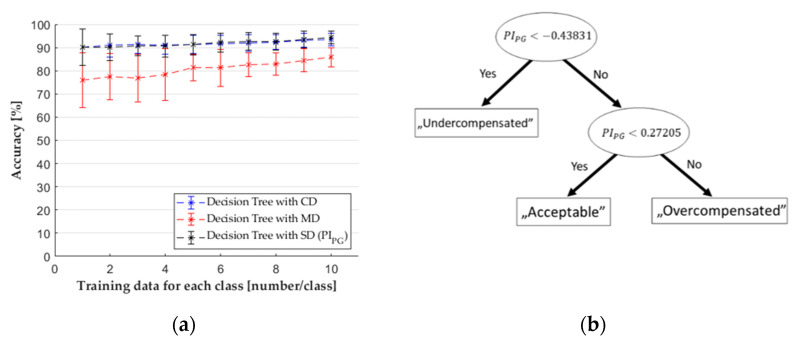
(**a**) The results of decision tree classifiers with different datasets; (**b**) a decision tree.

**Figure 11 sensors-22-02704-f011:**
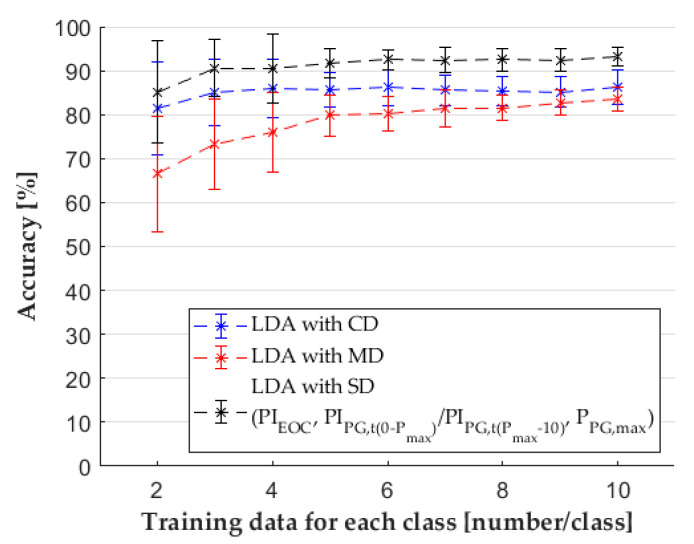
The results of the LDA classifier with different datasets.

**Figure 12 sensors-22-02704-f012:**
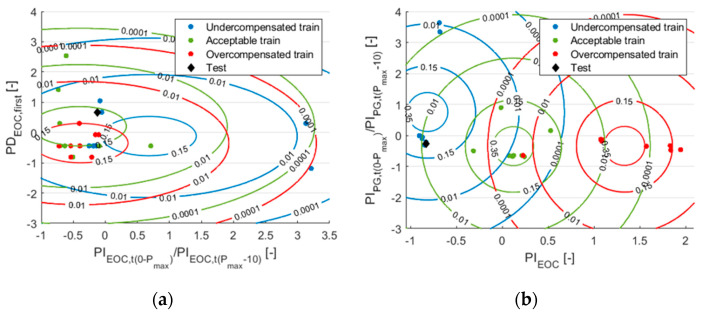
Training data and test data (from the uncompensated class) with distributions presented in two different dimension planes: (**a**) in two “worse” dimensions; (**b**) in the two “best” dimensions selected by the SFS method. Note that we used eight training samples per class.

**Figure 13 sensors-22-02704-f013:**
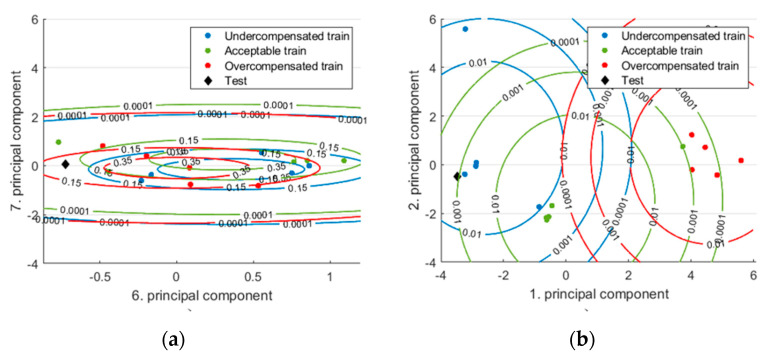
Training data and test data (from the undercompensated class) with distributions presented in two different dimension planes: (**a**) in dimensions created by the sixth and the seventh principal component; (**b**) in dimensions created by the first and the second principal component. Note that we used five training samples per class.

**Figure 14 sensors-22-02704-f014:**
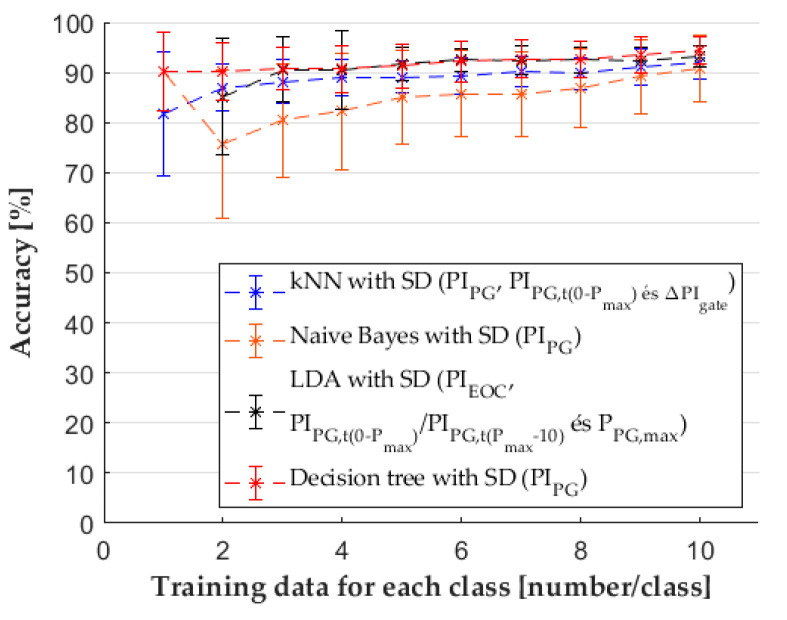
Comparison of classifiers with features chosen by the SFS method.

**Figure 15 sensors-22-02704-f015:**
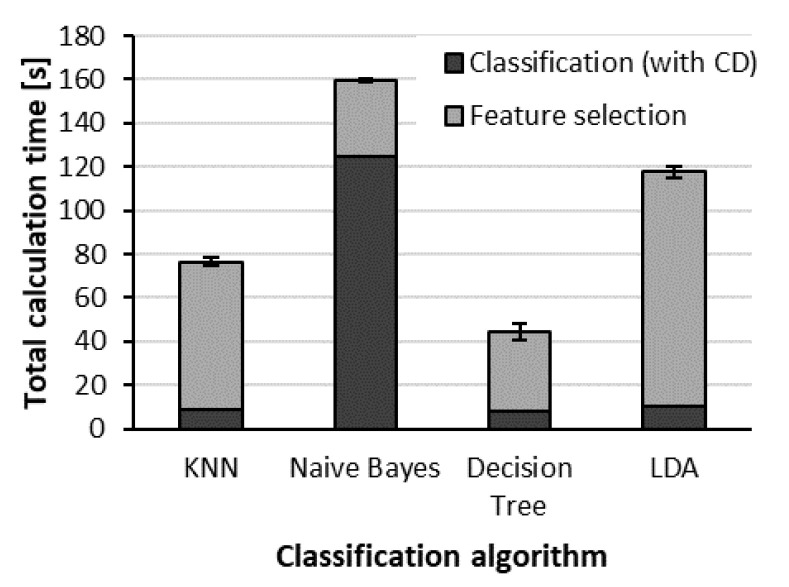
Calculation times for each classifier.

**Table 1 sensors-22-02704-t001:** The unchanged settings of injection molding.

Process Parameter	Value
Shot volume [cm^3^]	26
Screw rotation speed [m/min]	15
Back pressure [bar]	40
Decompression [cm^3^]	5
Injection velocity [cm^3^/s]	50
Switchover volume [cm^3^]	7
Injection pressure limit [bar]	1500
Clamping force [kN]	700
Cooling time [s]	18
Cycle time [s]	28
Melt temperature [°C]	225
Mold temperature [°C]	40

**Table 2 sensors-22-02704-t002:** Comparison of different classifiers and their performance with CD and the calculation time for feature selection.

Classifier	2 Samples/Class			10 Samples/Class			Calculation Time for the CD [s]	SFS Calculation Time [s]
	Average Performance [%]	Worst Case [%]	Best Case [%]	Average Performance [%]	Worst Case [%]	Best Case [%]		
kNN	77.77	50.54	88.04	85.14	73.13	93.12	8–10	64
Naïve Bayes	70.34	29.35	90.76	88.27	72.50	94.38	117	34–35
Decision Tree	90.97	75.54	97.28	93.61	85.63	99.38	8–10	34–35
LDA	81.42	45.11	94.57	86.26	77.50	95.63	8–10	101

## Data Availability

Not applicable.

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
