# Peer review of "Machine Learning in Injection Molding: An Industry 4.0 Method of Quality Prediction"

_sensors, 2022, doi:10.3390/s22072704_

Round 1
Reviewer 1 Report
In this article, four machine learning algorithms (kNN, naïve Bayes, linear discriminant analysis, and decision tree) are used and comparatively assessed in predicting the weight of injection molded parts.
The only original claim of this paper is that simple classifiers can be used to predict quality in injection molding even with few training data. However, this conclusion is affected by a major methodological flaw as explained in the following.
Out of 190 specimens used for training and testing only 1/5 is actually produced varying the holding pressure on 3 levels and the holding time on 12 levels (+1 reference level). The rest of the samples are replications and therefore are useful for assessing the noise of the process not for training.
Only 39 out of the 190 rows in the dataset contain information related to the effect of the process parameters on part quality (i.e. weight). This dataset is therefore too little to train any model with 19 features, in the complete dataset case, or even 8 features in the modified dataset case. No information is provided for the dimension of the third dataset.
Furthermore, the description of the mold and the molded parts is too vague. What geometry and dimensions do they have? What specific pressure sensors did you use? Those produced by Cavity Eye Ltd. are quite large to easily fit an injection mold in such a number (30). Can you provide a photo of the mold?
Author Response
Dear Reviewer,
Thank you very much for your comments on our manuscript and your advice. Our respond is attached as a pdf file.
Best regards,
The authors

Reviewer 2 Report
The authors present a solid industrial-driven article. They provided a comprehensive comparison between several different ML algorithms to predict the quality of an injection molding process. However, there are a few instances where we see potential for even further improvement:
- Although the discussion of different classes of machine learning is important, we don’t see the necessity to mention reinforcement learning explicitly because the authors did not use any method of that sort.
- An explicit description for Figure 2 a) is missing.
- The choice of features (defined in equation 1 to 10) is not motivated. It might be obvious for engineers that are very familiar with the field of injection molding. For others with different professional orientation, it is not so clear.
- Is there a specific reason that the band of acceptable products is relatively narrow and only includes around 21% of all produced products? Although the authors mentioned that they are deliberately chose that narrow band and what the implications are, they did not explain reason why. It would be preferable to state if that is an artificial setup for a proof-of-concept or the actual production performance.
- We are not sure what the index vector on the right side in Figure 5 means. It seems like that 6 indexes are associated with one element in the index table. We assume that it should be only one per element.
Author Response

(The authors gave the same response as above.)

Reviewer 3 Report
Dear Authors,
This research work was implemented on the Machine learning in injection moulding, industry 4.0 method of quality prediction, and I propose to improve the quality of the manuscript based on the comments below.
In the manuscript, the parameters of the injection moulding process are not clearly identified beyond the working condition and do not predict the machine's behaviour.
The manuscript omits the injection molding's remote location entirely.
Identify the machine's behaviour in comparison to the experimental and predicted ones, and compare the results with valid evidence and relevant literature work.
The final injection moulding outcome of the manuscript with appropriate application of the society product comparison of the existing product development is required.
Author Response
Dear Reviewer,
Thank you very much for your comments on our manuscript and your advice. Our response is attached as a pdf file.
Best regards,
The authors

Round 2
Reviewer 1 Report
The authors' reply to my first comment (i.e. "the noise of the technology was so small that we could not produce a different class of product with the same settings") prove that they erroneously used 151 out of 190 specimens to train the models. This is a methodological error. Only 39 out of the 190 rows in the dataset contain information related to the effect of the process parameters on part quality (i.e. weight). This dataset is therefore too little to train any model with 19 features, in the complete dataset case, or even 8 features in the modified dataset case.
The authors justify this choice by stating that “the general rule is that sample size must exceed the number of features". Again, this is not true. The generalization theory, based on the concept of the Vapnik-Chervonenkis dimension (dVC), studies the cases in which it is possible to generalize out of sample what we find in sample. The takeaway concept is that learning is feasible in a probabilistic way. If we are able to deal with the approximation-generalization tradeoff, we can say with high probability that the generalization error is small. As a general rule of thumb, the number of data points (N) required to ensure a good generalization bound needs to be an order of magnitude greater than the Vapnik-Chervonenkis dimension: (? ≥ 10 ⋅ ???), which is usually approximated with the number of model parameters (model complexity). See the extensive work of prof. Peter Bartlett (EECS and Statistics, UC Berkeley) on this topic.
Regarding my last remark on the sensors, pressure transducers placed behind ejectors pins are old technology. Compared to flush-mounted piezoelectric transducers, they are less accurate and have slower response to pressure variations due to friction and non-perfect stiffness of the pins. It is curious that in a journal called “Sensors” information about sensors is not included in the manuscript for simplicity or confidential reasons.
Author Response
Dear Sir or Madam,
Thank you very much for your comments on our manuscript and your advice. We uploaded our respond to your points.
Regards,
Authors

Reviewer 3 Report
Dear Authors,
You have incorporated all of my suggestions to improve the quality, except for one of the references you have mentioned in the author’s reply for a recent review article that summarises progress in this area (S. K. Selvaraj, A. Raj, R. R. Mahadevan, U. Chadha, V. Paramasivam). Machine Learning Models in Injection Molding Machines: A Review Hindawi 2022, Article ID 1949061, 28 pages (doi:10.1155/2022/1949061). Please include the above references in the reference section, as only most readers are aware of the most recent changes.
Author Response
Dear Sir or Madam!
Thank you again for your recommendations, we added the mentioned article to the manuscript.
Best regards
Authors